# Extracellular Vesicles and Exosomes: Novel Insights and Perspectives on Lung Cancer from Early Detection to Targeted Treatment

**DOI:** 10.3390/biomedicines12010123

**Published:** 2024-01-07

**Authors:** Sana Rahimian, Hossein Najafi, Baran Afzali, Mohammad Doroudian

**Affiliations:** Department of Cell and Molecular Sciences, Faculty of Biological Sciences, Kharazmi University, Tehran 14911-15719, Iran; sanarahimian@gmail.com (S.R.); hosseinnajafi7979@gmail.com (H.N.); baranafzali2001@gmail.com (B.A.)

**Keywords:** exosomes, circulating exosomes, lung cancer, NSCLC, diagnosis, biomarkers, non-invasive therapeutics, theranostics, targeted drug delivery

## Abstract

Lung cancer demands innovative approaches for early detection and targeted treatment. In addressing this urgent need, exosomes play a pivotal role in revolutionizing both the early detection and targeted treatment of lung cancer. Their remarkable capacity to encapsulate a diverse range of biomolecules, traverse biological barriers, and be engineered with specific targeting molecules makes them highly promising for both diagnostic markers and precise drug delivery to cancer cells. Furthermore, an in-depth analysis of exosomal content and biogenesis offers crucial insights into the molecular profile of lung tumors. This knowledge holds significant potential for the development of targeted therapies and innovative diagnostic strategies for cancer. Despite notable progress in this field, challenges in standardization and cargo loading persist. Collaborative research efforts are imperative to maximize the potential of exosomes and advance the field of precision medicine for the benefit of lung cancer patients.

## 1. Introduction

Cancer, as stated by the World Health Organization (WHO), stands as a leading cause of mortality worldwide [1]. Specifically, lung cancer is among the most reported cancer-related deaths globally [2,3], constituting 11.6% of new cancer diagnoses and accounting for 19.8% of cancer-related fatalities [4,5].

Regrettably, despite a decline in lung cancer mortality rates, a significant proportion of patients continue to receive diagnoses in advanced or metastatic stages, resulting in poor prognoses [6]. Advanced stages pose greater therapeutic challenges and exhibit a heightened propensity for developing resistance to treatment. Various treatment modalities have been employed in lung cancer care, each with its own advantages and drawbacks. Surgery remains a viable option for early-stage cases [7]. Chemotherapy, radiotherapy, and immunotherapy, either individually or in combination, are typical approaches for both early and advanced stages [8,9,10]. The differential factor lies in the responsiveness of lung cancer cells to these treatments, with a higher sensitivity observed during the early stages.

Hence, the study of tumorigenesis and therapeutic strategies against cancers, particularly lung cancer, is underscored. Recent research has yielded fresh insights into lung cancer biology, yielding notable progress in the realm of targeted therapies focused on novel biomarkers. These include molecular therapies targeting the epidermal growth factor receptor (EGFR), anaplastic lymphoma kinase (ALK), proto-oncogene B-Raf (BRAF), and immunotherapies employing checkpoint inhibitors against the programmed cell death 1 (PD-1) and programmed death-ligand 1 (PD-L1) pathways [8,11,12]. Nonetheless, lung cancer still lags behind with one of the lowest 5-year survival rates, standing at a mere 18%, among all cancer types [9].

Extracellular vesicles (EVs), which are a collective term covering exosomes, microvesicles (MVs), microparticles, ectosomes, oncosomes, and apoptotic bodies, were first discovered by Pan and Johnson during the maturation of sheep reticulocytes in 1983 [13]. EVs are defined as heterogeneous entities of phospholipid bilayer membrane-bound vesicles without any means of replication [14]. They were initially thought to be “cellular dust” or served as a mechanism by which cells actively dispose of their own waste [15]. In 1996, exosomes were found to play a role in intercellular communication [16]. Due to the heterogeneity between and within exosome types and the overlap in characteristics between exosomes and other EVs, it is difficult to define exosomes in a way that distinguishes them from other EVs. Over the past few years, it has been discovered that exosomes, which contain proteins and miRNAs with biological effects, have the ability to specifically transport and deliver these bioactive cargoes to tumor cells [15].

Exosomes, ranging from 30 to 150 nanometers in diameter, emerge as crucial biological nano-scale lipid bilayer vesicles [17,18]. These vesicles are secreted by various cell types, including dendritic cells, macrophages, B cells, T cells, mesenchymal stem cells, endothelial cells, epithelial cells, and several cancer cells. They float within a sucrose density gradient solution at a density of 1.13–1.19 g mL^−1^ and carry an array of biomolecules, such as proteins (either membrane-bound or encapsulated within the vesicle), RNA (comprising coding mRNA or diverse non-coding RNAs), DNA (both double-stranded and single-stranded), as well as glycans [16,19,20]. They have been reported in all biological fluids, and the composition of the complex cargo of exosomes is readily accessible via the sampling of biological fluids (liquid biopsies) [21,22]. Cumulative evidence has revealed that exosomes play an irreplaceable role in prognostic, diagnostic, and even therapeutic aspects [21,23]. Nanotechnology has been employed for drug delivery for increasing the bioavailability of therapeutic agents. Unfortunately, drug nanoformulations often lead to toxicity and are usually rapidly cleared through the mononuclear phagocytic system (MPS) [24,25].

EVs can be categorized, based on their size, into microvesicles (MVs; 100–1000 nm) and exosomes (EXOs; 30–150 nm). They play pivotal roles in immune responses, angiogenesis, neuronal regeneration, anti-inflammation, coagulation, and in the intracellular degradation pathways [14,26].

Exosomes’ unique properties make them promising tools for therapeutically targeting diseases, including neurodegenerative conditions and various cancers [27]. They have shown promise as non-invasive biomarkers for chronic pain mechanisms [28] and as potential carriers for biomarkers and drugs in conditions such as human immunodeficiency virus (HIV) [29]. They can cross biological barriers, including the blood–brain barrier, and have neuroprotective and tissue repair effects [30,31]. It has been demonstrated that exosomes have potential applications in a wide range of fields, from oncology (lung, liver, pancreatic, colorectal, gastric, kidney, bladder, prostate, breast, ovarian, cervical, head and neck, thyroid, glioma, melanoma, and hematological malignancies) to neurodegenerative disorders, including Alzheimer’s disease, Parkinson’s disease, Huntington’s diseases, and amyotrophic lateral sclerosis (ALS), mental health conditions, cardiovascular diseases, diabetes, and inflammatory/autoimmune disorders [27].

Numerous studies have investigated exosomes from diverse angles in the context of lung cancer. These encompass experimental techniques and isolation methods, detailed characterization, insights into biogenesis and secretion, understanding molecular composition and functions, communication within the tumor environment, impact on tumor progression, and the potential clinical applications in cancer treatment, among other varied perspectives.

## 2. The Biogenesis of Exosomes

Exosomes are small vesicles generated from late endosomes, which are formed via the inward budding of the multivesicular body (MVB) membrane. They are different from other vesicles called MVs, which are larger and generated via outward budding from the plasma membrane [32]. Exosomes have an approximate size range from 30 nm to 150 nm [17,18] and contain proteins, mRNAs, microRNAs, and lipids. They are derived from various cellular sources and have biomarkers such as TSG101, CHMP2A, RAB11B, CD63, and CD81 [33]. Exosomes play a crucial role in cell-to-cell communication and are involved in various biological processes and diseases [34]. They can escape clearance by the immune system and have a longer circulation time compared to other vesicles [32]. The formation of exosomes involves the invagination of late endosomal membranes, resulting in the formation of intraluminal vesicles (ILVs) within MVBs [35]. ILVs are eventually secreted as exosomes through fusion of MVBs with the plasma membrane [36]. The content of exosomes depends on the cell type they originate from and can include proteins and components from various cellular compartments [33].

Exosomes have a distinct shape, appearing biconcave or cup-like when dried and spheroid when observed under a microscope [32]. Exosomes can form early-sorting endosomes (ESEs) de novo or merge with existing ESEs. The trans-Golgi network and endoplasmic reticulum contribute to ESE formation. ESEs mature into late-sorting endosomes (LSEs) and eventually generate MVBs, also known as multivesicular endosomes [20]. During maturation, the content of endosomes becomes acidic, and markers such as Rab5 are replaced by Rab7, Rab9, and the mannose-6-phosphate receptor to indicate late endosomes [35]. The influence of specific markers or cargo on these pathways is still unclear [37].

Several mechanisms contribute to exosome generation. The endosomal sorting complexes required for transport (ESCRT-0, -I, -II, and -III) and the ATPase Vps4 complex are important in this process [38]. SNARE proteins and RAB GTPases are also involved in exosome secretion [39,40]. Even when key subunits of ESCRTs are depleted, ESCRT-independent ILV biogenesis can still occur [41]. Tetraspanins and lipids play significant roles in exosome generation and release [42]. Identifying molecules that regulate exosome biogenesis could lead to therapeutic targets for controlling exosome secretion and modulating related signaling pathways in conditions such as metastasis and inflammation [43,44]. Cholesterol treatment in hepatoma cells reduces MVBs and increases exosome secretion with the ability to induce specific immune responses. Statins, known for lowering cholesterol, have been shown to reduce exosome release, suggesting their potential as therapeutic agents to control exosome production in specific cells [42].

Exosome biogenesis and release are influenced by various molecules, including components of the ESCRT machinery, Rab GTPases, tetraspanins, and the adaptor protein syntenin. The ESCRT mechanism comprises four separate protein complexes (ESCRT-0 through III) that work together to facilitate the formation of MVBs, vesicle budding, and sorting of protein cargo. Ubiquitinated proteins are recognized and sequestered to specific domains of the endosomal membrane by the ubiquitin-binding subunits of ESCRT-0, initiating the process. ESCRT-I and ESCRT-II complexes interact with ESCRT-III to promote budding. The ESCRT-III complex then separates from the MVB membrane, aided by the Vps4 sorting protein, after cleaving the buds to form ILVs [32,45].

In a study using HeLa cells, an RNAi screen targeting twenty-three ESCRT and ESCRT-associated proteins identified seven proteins that impacted exosome secretion. The depletion of ESCRT-0 proteins (Hrs and TSG101) and the ESCRT-I protein STAM1 reduced exosome secretion, while the knockdown of ESCRT-III and its associated proteins (CHMP4C, VPS4B, VTA1, and ALIX) increased exosome secretion. Further analysis verified the role of these proteins, with Hrs, TSG101, and STAM1 depletion decreasing exosome secretion and VPS4B knockdown increasing it. ALIX depletion appeared to affect the protein composition of exosomes rather than secretion, suggesting it may impact cargo loading and the types of MVBs destined for secretion [46,47].

The interaction between syntenin and ALIX is important for sorting syndecans, membrane proteins with heparan sulfate chains, into exosomes. Syndecan sorting and ILV formation are facilitated by syntenin, which binds to both syndecans and ALIX [47,48]. Heparanases trim the heparan sulfate chains, promoting the formation of syndecan clusters that enhance binding to syntenin. Heparanase also stimulates the sorting of CD63, indicating a potential relationship between the sorting of these two molecules [47,49]. The syndecan–syntenin–ALIX mechanism is estimated to control around 50% of secreted vesicles in MCF-7 cells, suggesting the involvement of different sorting mechanisms in exosomal molecule sorting [47,50]. These findings suggest that ESCRT function plays a role in exosomal biogenesis.

Cellular metabolic status, including ceramide metabolism, ER stress, autophagy, and intracellular calcium levels, can impact exosome production. Adiponectin, a protein secreted by adipocytes, can stimulate exosome biogenesis in certain cell types [38,51]. Glucagon has been found to regulate exosome production in endothelial cells in adipose tissues [38,52]. The interaction between the exosome biogenesis pathway and the other molecular pathways involved in intracellular vesicle trafficking can complicate the interpretation of functional studies. Different cell types, culture conditions, and cell health can also affect the regulation of exosome biogenesis [20,53]. Inconsistencies in identifying regulatory elements may arise from variations in the methods used for exosome production, enrichment, and concentration [20,54].

The specific case of exosome biogenesis in human syncytiotrophoblasts has been studied at the ultrastructural level. Certain molecules, such as MICA/B, ULBP 1–5, FasL, TRAIL, and PD-L1, are present on the membranes of MVBs and nanosized vesicles within MVBs, but not on the plasma membrane of syncytiotrophoblasts. These molecules may be sorted from the Golgi apparatus to the MVBs. MICA and MICB are expressed on the apical membrane of syncytiotrophoblasts and within exosomes contained within MVBs [35,55].

ESCRTs are not only involved in exosome release but also play a role in packaging biomolecules into exosomes [56]. Heparanase modulates the syndecan–syntenin–ALIX pathway, leading to endosomal membrane budding and subsequent exosome formation by trimming heparan sulfate chains on syndecans [49,56]. However, previous research indicated that cargo molecules within exosomes are segregated into distinct subdomains on the endosomal membrane [56,57]. Additionally, the transfer of exosome-associated domains into the endosomal lumen does not rely on ESCRT function but is induced by sphingolipid ceramide. Purified exosomes contain ceramides, and the inhibition of neutral sphingomyelinases reduces exosome release, suggesting the regulatory role of lipids in exosome secretion [56].

Protein sorting within MVBs, the precursors of exosomes, can occur through the ESCRT-dependent and ESCRT-independent pathways. These pathways may work in synergy, and different subpopulations of exosomes may depend on distinct machinery [47,58] (Figure 1).

## 3. The Isolation of Exosomes

One of the main challenges in exosome research is the lack of an efficient and standardized isolation strategy for specific exosome subpopulations, leading to heterogeneity in their size and molecular contents. However, advances in nanotechnologies and microfluidics have shown promise in incorporating microfluidics into exosome isolation, offering potential tools for future research [59].

Exosomes, although playing a crucial role in early detection and treatment, present challenges due to their small size, low density, and similarity to other EVs in body fluids [60]. Separating the exosomes from cell debris and other EVs is essential, and different isolation strategies based on their size and affinity have been developed [61]. These include ultracentrifugation, size-based techniques, immunoaffinity capture, EV precipitation, and microfluidics-based techniques. Each method has its advantages and disadvantages, and the appropriate choice depends on the research purpose [62,63]. Microfluidics, in particular, offers an integrated platform with properties such as high purity, high throughput, and low sample volumes [59].

While traditional methods, like ultracentrifugation, are widely used, they have drawbacks, such as large sample consumption, potential damage to exosomes, low purity, and lengthy procedures. Common exosome isolation technologies, such as ultrafiltration, immunoaffinity, and ultracentrifugation, can be expensive and result in low isolation efficiency, sample loss, and contamination [59,64].

In summary, the development of efficient and standardized isolation strategies for specific exosome subpopulations remains a challenge. However, advancements in microfluidics and other techniques offer promising avenues for future research, with microfluidics showing potential due to its integrated platform and desirable properties [59]. The choice of isolation method should consider the research purpose, and while traditional methods are widely used, they have limitations that hinder meeting the increasing demands of scientific research [63].

## 4. Exosomes: Effective Drug Delivery Vehicles

The development of drug delivery systems (DDSs) with enhanced therapeutic effectiveness has garnered significant attention [65,66]. Nanoscale drug carriers offer advantages over traditional chemotherapeutic agents, including targeted accumulation in tumors and reduced toxicity in normal cells [67]. Active targeting can be achieved by binding nanocarriers to molecules that bind to overexpressed antigens [68]. While most drug delivery vehicles are chemically synthesized using lipids or lipid-like molecules, natural nanoparticles derived from bacteria, viruses, red blood cells, and lymphocytes have emerged as potential alternatives, as they exhibit advantages such as evasion of the host immune system and efficient cellular entry [69].

Exosomes, a type of natural nanoparticle, have been investigated as direct therapeutic approaches for treating diseases or injuries [30]. They possess the ability to transfer proteins and nucleic acids between cells, making them suitable for drug delivery. Exosomes offer benefits, such as tissue specificity, safety, and stability, as they can deliver cargo across plasma membranes into the correct cellular compartments to exert a functional response [69,70]. They also protect their cargo from clearance or damage by complement fixation or macrophages due to their double-layered membrane and nanoscale size, thereby prolonging circulation and improving biological activity [71]. Additionally, exosomes display intrinsic homing capabilities, enabling them to specifically target tissues [72].

Although the field of exosome-based therapeutics is still in its early stages, progress has been made in engineering exosomes to display specific proteins, incorporate nucleic acid and protein cargo, and load therapeutic agents [73]. The size and stability of exosomes make them favorable for drug delivery systems, as they exhibit reduced aggregation and precipitation. For example, milk-derived exosomes can be stored for periods of time without causing a loss of activity. Overall, exosomal drug delivery holds the potential to enhance drug efficacy by prolonging circulation times and facilitating cellular uptake [74].

## 5. Approaches for Drug Loading on Exosomes

Methods for loading cargo into exosomes can be categorized into cell-based and non-cell-based approaches. In the cell-based approach, cargo is delivered to donor cells, which then package it into EVs for secretion and therapeutic use. The non-cell-based approach involves directly loading drugs into isolated EVs using techniques like electroporation, extrusion, sonication, and bio-conjugation. However, the loading efficiencies of these methods are not always desirable. The drug encapsulation process can occur through postloading, preloading, or fusion methods, and active loading and passive methods are employed to overcome the challenges of incorporating drugs into the lipid bilayer membrane of exosomes [75,76].

To enhance the therapeutic potential of exosomes, various targeting strategies are employed. These strategies can involve cellular modification through engineering techniques, altering the surface expression molecules and cargo of exosomes [73]. Modifying exosomal tetraspanin complexes can improve target cell selection and enhance tissue and cell type-specific targeting [77]. Lentiviral vectors can be utilized to create exosomes with modified membrane proteins for in vivo and cell uptake studies [78]. Alternatively, direct modifications can be achieved through passive loading techniques that leverage spontaneous membrane interactions or physical methods that temporarily disrupt the integrity of the membranes to facilitate cargo loading into exosomes [75].

## 6. Lung Cancer and the Role of Exosomes in Its Diagnosis and Treatment

Lung cancer is classified into two primary categories: small cell lung cancer (SCLC) and non-small cell lung cancer (NSCLC) [79].

SCLC is an extremely aggressive neuroendocrine carcinoma. The prognosis for SCLC patients is extremely poor, with a 5-year survival rate of less than 5% and an average overall survival period of only 2–4 months for untreated patients. SCLC represents approximately 15% of all lung cancer cases [80,81].

In contrast, NSCLC comprises approximately 85% of all lung cancer cases [82]. Histologically, NSCLC is classified into different types, namely adenocarcinoma, large-cell carcinoma, and squamous cell carcinoma. The stages of this disease are further categorized based on specific criteria [3,79]. Adenocarcinoma represents 38.5% of all lung cancer cases, while squamous-cell carcinoma accounts for 20%, and large-cell carcinoma accounts for 2.9% [4] (Figure 2).

### 6.1. Role of Exosomes in Lung Cancer Diagnosis

Exosomes are emerging as valuable candidates for both diagnosing and predicting lung cancer due to their detectability in various bodily fluids, such as plasma, serum, and alveolar lavage fluid. Traditional methods, like low-dose computed tomography (LDCT), for lung cancer screening can result in cumulative radiation exposure and side effects. Tissue sample analysis, the gold standard, is hindered by sampling bias and invasiveness. In contrast, exosomes provide a non-invasive liquid biopsy option with the potential for early cancer detection. They carry unique proteins, receptors, signaling molecules, and lipids that can be compared with healthy controls to identify cellular abnormalities. Exosomes hold promise as diagnostic tools for chronic diseases and are particularly valuable in biomarker research due to their stability, cargo of proteins and RNA, and ability to present antigens [87,88,89].

There are several clinical trials that have been conducted on lung cancer diagnosis via exosome detection in clinicaltrials.gov/, (accessed on 27 November 2023) (Table 1). Nano-sized exosomes in circulation offer components, like proteins, miRNAs, and lncRNAs, for cancer diagnosis and monitoring. They contain mRNA, miRNAs, dsDNA, and proteins, making them valuable predictive and prognostic biomarkers for different tumor types. Their role in intercellular communication and protection of membrane bilayers and contents from degradation adds to their potential [88,89,90].

Researchers have identified exosomal miRNAs as promising biomarkers for early lung cancer diagnosis due to their stability, accessibility, and specificity [87]. One study highlighted miR-1268b and miR-6075 as promising serum markers for lung cancer diagnosis. Elevated levels of these biomarkers were detected even during early-stage lung cancer, indicating their potential as an early diagnostic indicator. When combined, miR-1268b and miR-6075 showed a high accuracy rate in distinguishing lung cancer patients from non-cancerous individuals, offering high sensitivity and specificity [91]. Moreover, research has indicated that exosomal miR-205 expression can differentiate between squamous and non-squamous NSCLCs, even in poorly differentiated tumors [92].

Another study discovered heightened miR-96 levels in lung cancer patients, notably in high-grade cancers, observed in tissues, serum, and isolated exosomes. This finding suggested its potential as a diagnostic biomarker for lung cancer. Elevated exosomal miR-96 correlated with increased lung cancer risk, higher stages, and metastatic tumors in the lymph nodes, indicating its involvement in cancer progression. It holds promise for diagnosing and predicting outcomes in NSCLC. The detection of miR-96 in serum could facilitate early identification, accurately assess cancer aggressiveness, and predict patient survival [93]. A study that focused on miR-451a in NSCLC patients post-surgery found it to be a promising biomarker for diagnosing cancer and predicting recurrence and prognosis across different disease stages (I–III). Higher levels of exosomal miR-451a correlated with lymph node metastasis, vascular invasion, and disease stage. This study hinted that these miRNAs in the plasma likely originate from the tumor tissue itself, supported by the correlation observed between the levels of exosomal miR-451a and its expression in the primary tumor tissues of the same patients [94].

Moreover, deregulation in the expression of miR-21, miR-143, and miR-181a has been correlated with clinical pathology and patient prognosis in NSCLC, indicating their usefulness as diagnostic or prognostic markers [95]. Additionally, studies have also linked miR-21 and miR-4257 expression with recurrence and poor survival in lung cancer cases [96]. Exosomal miRNAs, such as miR-181-5p, miR-30a-3p, miR-30e-3p, and miR-320b, have also been identified as biomarkers in NSCLC to differentiate between adenocarcinoma and squamous-cell carcinoma [97,98].

A pivotal breakthrough in the realm of lung cancer biomarkers took place in 2009 with the discovery of a group of 12 miRNAs (miR-17-3p, miR-21, miR-106a, miR-146, miR-155, miR-199, miR-192, miR-203, miR-205, miR-210, miR-212, and miR-214). These miRNAs effectively distinguished NSCLC patients from their healthy counterparts. Remarkably, these miRNAs were not only detectable in lung cancer tissues but also in circulating exosomes, suggesting their potential as diagnostic indicators. Moreover, the levels of these miRNAs in the exosomes from NSCLC patients exhibited a substantial elevation when compared to those from healthy donors. This discovery held immense promise for more precise and timely diagnoses of lung cancer [99,100]. Further research has expanded these findings, leading to the development of miRNA profiles for lung cancer screening and diagnosis. A study introduced two miRNA profiles based on four miRNAs for screening and six miRNAs for diagnosis [101]. These profiles demonstrated high sensitivity and specificity, indicating their potential utility in identifying lung adenocarcinomas. Another study established a miRNA profile consisting of four miRNAs with high diagnostic accuracy for NSCLC, further highlighting the potential of exosomal miRNAs as diagnostic biomarkers [90,98].

Notably, the association of specific miRNAs with prognostic outcomes in NSCLC reveals their potential as valuable indicators for disease progression and treatment response. For instance, the downregulation of miR-503 in NSCLC tissues associates with advanced tumor stages and a poor prognosis, hinting at its potential as a prognostic indicator [102]. Additionally, exosomal miRNAs like miR-146-5p, miR-23b-3p, miR-10b-5p, and miR-21-5p have shown connections to survival rates and treatment responses in NSCLC. Combinations of these miRNAs and clinical variables enhance predictive models [98,103]. Moreover, miR-1246, miR-208a, and miR-100-5p have been linked to tumor proliferation, radiotherapy resistance, and mediating resistance to cisplatin, underscoring the potential of exosomal miRNAs in understanding and foreseeing treatment resistance in NSCLC [104,105,106].

Tumor-derived exosomal miRNAs and proteins play intricate roles in lung cancer metastasis, influencing various pathways and cellular processes. Exosomal miR-619-5p promotes metastasis by targeting the tumor suppressor RCAN1.4. miRNAs such as miR-433 and miR-1260b interact with the Wnt/β-catenin pathway, influencing NSCLC progression and metastasis [107]. Several trials using exosomes with miRNA cargo for lung cancer diagnosis are outlined in Table 2.

**Table 2 biomedicines-12-00123-t002:** Clinical use of exosomes as biomarkers.

	Molecules Detected/Cargo Used	Sample	Function	Reference
1	miR-375, miR-429, miR-203a-3p, miR-141-3p, miR-205-5p, miR-483-5p, miR-200c-3p, miR-200b-3p, and miR-200a-3p	Pleural effusion	Diagnostic biomarkers for distinguishing lung adenocarcinoma over tuberculosis and other benign lesions	[108]
2	miR-30a-3p, miR-100, miR-139-5p, miR-151a-5p, miR-379, miR-154-3p, miR-200b-5p, miR-378a, and miR-629	Plasma	Diagnosis	[101]
3	miR-9-3p, miR-205-5p, miR-210-5p, and miR-1269a	Serum	Diagnosis	[109]
4	miR-34b, miR-125b, miR-200b, miR-203, miR-429, and miR-205	Serum	Diagnosis (including early stage)	[110]
5	miR-19a, miR-19b, miR-20a, miR-30b, and miR-205	Plasma	Diagnostic biomarkers of lung squamous-cell carcinoma (SCC)	[111]
6	miR-328-3p, miR-423-3p, and miR-574-5p	Plasma	Bone metastasis detection	[112]
7	miR-10b-5p, miR-21-5p, and miR-23b-3p	Plasma	Prognostic biomarkers for overall survival	[103]
8	miR-19-3p, miR-221-3p, and miR-21-5p	Plasma	Diagnostic biomarkers for lung adenocarcinoma	[113]
9	miR-21 and miR-4257	Plasma	Prognostic biomarkers for disease-free survival (recurrence)	[114]
10	miR-20b-5p and miR-3187-5p	Serum	Diagnosis (including early stage)	[115]
11	miR-182 and miR-210	Pleural effusion	Diagnostic biomarker for malignant pleural effusion (lung adenocarcinoma) over benign (non-neoplastic) pleural effusion	[116]
12	miR-320a, miR-622	Plasma	Metastasis detection	[117]
13	miR-200b and miR-205-5p	Pleural effusion	Diagnostic biomarker for lung cancer	[118]
14	miR-29a-3p and miR-150-5p	Plasma	predictive biomarker for unexpected responses to radiation therapy	[119]
15	miR-200	Pleural effusion	Diagnostic biomarker for malignant pleural effusion (lung adenocarcinoma) over benign (non-neoplastic) pleural effusion	[120]
16	miR-5684	Serum	Diagnosis	[121]
17	miR-125b-5p	Serum	Diagnosis Prognosis: metastasis detection and survival Therapy monitoring	[121]
18	miR-23b-3p	Serum	Diagnosis Prognosis: tumor size, depth of invasion, liver metastasis, and TNM stage	[122]
19	miR-21	Serum	Diagnosis (including versus benign and inflammatory lung diseases)	[123]
20	miRNA-205	Urine and saliva	Diagnosis	[124]
21	miR-146a-5p	Serum	Prognostic biomarker for recurrence	[125]
22	miR-1268b and miR-6075	Serum	Early detection of resectable lung cancer	[91]
23	miR-96	Serum	Early detection of lung cancer	[93]
24	miR-451a	Plasma	Diagnostic and prognostic: early detection, lung cancer aggressiveness, and patient survival	[94]
25	miR-181	Tissue	Prognostic NSCLC patients with stage I, II, or III cancer	[95]
26	miR-503	Tissue	Prognostic biomarker in NSCLC.	[102]
27	miR-181-5p, miR-30a-3p, miR-30e-3p, and miR-320b	Plasma	Differentiating adenocarcinoma and squamous-cell carcinoma in NSCLC	[98]
28	miR-17-3p, miR-21, miR-106a, miR-146, miR-155, miR-199, miR-192, miR-203, miR-205, miR-210, miR-212, and miR-214	Plasma	Diagnostic and prognostic biomarkers of adenocarcinoma	[99,100]
29	miR-146-5p, miR-23b-3p, miR-10b-5p, and miR-21-5p		Predicting survival rates and treatment responses in NSCLC	[124,126]
30	miR-619-5p, miR-433, and miR-1260b		Progression and metastasis detection	[127]

Exosomal proteins are being explored as potential biomarkers for NSCLC, similar to miRNAs. Exosomal proteins present a promising avenue in the quest for effective biomarkers in NSCLC diagnosis and prognosis, akin to the role played by miRNAs. Among these proteins, EGFR, KRAS, EMMPRIN, claudins, and RAB family proteins have emerged as key players associated with lung cancer [90,128]. Notably, exosomal EGFR, a membrane protein, has been notably prevalent in NSCLC exosomes. Proteomic analyses have unearthed a repertoire of differentially expressed proteins in NSCLC exosomes in contrast to their normal counterparts, highlighting their potential diagnostic value [129]. Additionally, specific markers, such as CD91 and CD317, have been proposed for NSCLC detection. Beyond diagnosis, the prognostic landscape has been explored, with membrane-bound proteins like NYESO-1 and PLAP linked to poorer outcomes. Excitingly, multi-marker models leveraging exosomal proteins have shown substantial promise in effectively discerning lung cancer patients from their controls. Notably, CD151, CD171, and tetraspanin 8 were identified as strong markers for differentiating lung cancer histological subtypes [130].

### 6.2. Role of Exosomes in Lung Cancer Treatment

Exosomes have shown great potential for revolutionizing lung cancer treatment. They serve as nanocarriers, offering advantages over traditional drugs by providing high payloads, precisely targeting tumors, and enhancing treatment effectiveness. Studies have indicated that exosomes induce G2 phase cell cycle arrest, apoptosis, and DNA damage, suggesting preferential toxicity to tumor cells. Moreover, exosomes derived from antigen-stimulated cells show their potential for vaccination [126,131,132,133].

Exosomes have gained prominence in immune regulation and hold promise as tools for cancer immunotherapy. Their capacity to present tumor antigens to immune cells positions them as potential solutions for addressing the immunogenicity challenge in cancer treatment. Tumor-derived exosomes loaded with antigens have been found to stimulate immune responses against tumors. Many investigations have been conducted on exosomes released by activated antigen-presenting cells (APCs), like dendritic cells (DCs), macrophages, T cells, and B cells [3,127,134,135,136]. Immune cells primed with antigens can package cancer cell components into exosomes, further promoting immune responses. In mouse models, exosomes secreted by DCs that are exposed to tumor antigens have shown promise in eradicating tumors. They induced tumor-sensitized T cells to secrete high levels of interferon-γ (IFN-γ), a cytokine that is important for immune responses [137].

Lung cancer cells use exosomes to communicate with immune cells, creating an environment that suppresses the immune response and supports tumor growth. Understanding this form of communication paves the way for developing targeted lung cancer immunotherapies by influencing the interactions mediated by exosomes. Lung cancer-derived exosomes carry specific surface markers that immune cells, particularly macrophages, take in. Consequently, these exosomes induce alterations in the macrophages’ characteristics, fostering an environment that promotes tumor growth and suppresses the immune response within the tumor microenvironment [138].

In a study, exosomes derived from lung tumor cells engineered with the CD40L gene (CD40L-EXO) were proven to be more effective in activating DCs and stimulating anti-tumor T cell immunity compared to standard exosomes. In lung cancer, DCs often transition from activating the immune system to fostering tolerance. CD40L-EXO demonstrated the ability to activate DCs, potentially reversing their tolerance status. When combined with the ability of heat-stressed tumor cell-derived exosomes to attract DCs to the tumor site, the use of CD40L-EXO appears to be promising for lung cancer treatment [139].

Exosomes released by NSCLC cells carry program death-ligand 1 (PD-L1) on their surface, playing a vital role in supporting tumor growth by hampering the function of CD8+ T cells, crucial for anti-tumor immunity, and triggering cell death in these T cells. These exosomes essentially aid the evasion of the immune system in NSCLC. Understanding the significance of PD-L1-positive exosomes is crucial in maximizing the effectiveness of PD-1/PD-L1 immunotherapy. Targeting these exosomes could offer a new approach to bolster the immune response against NSCLC, potentially improving the outcomes of immunotherapy treatments [140].

Tumor-derived exosomes possess immunosuppressive effects by hindering the maturation and migration of DCs, as well as the secretion of pro-inflammatory factors. The interaction between tumor-derived exosomes and DCs results in the upregulation of PD-L1, an immune checkpoint protein, which ultimately leads to the inhibition of the immune response. Blocking PD-L1 has shown potential in partially restoring the suppressed immune functions caused by exosome-treated DCs. This highlights the importance of considering both the cell surface and exosome presentation of PD-L1 in therapeutic strategies in combination with current immune checkpoint therapies [10,141,142].

In recent studies, the intricate connection between exosomes and immune responses in the context of cancer has been a subject of intense research [140,142,143]. Exosomes have demonstrated a significant influence on immune modulation and potential applications in cancer immunotherapy [144,145]. However, further research is needed to fully understand their mechanisms and optimize their use in clinical settings.

Building on their potential in cancer treatment, exosomes derived from tumor cells and innate immune cells have been explored for their use in vaccine development. These exosomes carry molecules such as MHC and costimulatory molecules that enhance immune cell responses against cancer. Exosome-based vaccines have exhibited positive outcomes against infectious diseases and are being considered for cancer immunotherapies [146].

In advanced NSCLC, a study revealed a significant decline in natural killer cell (NK cell) function, particularly in NKp30 expression, suggesting impairment. Soluble BAG6 levels were associated with weakened NK cell function in this context. Higher MHC-II and IFN-γ correlated with increased NK cell activity, particularly related to NKp30. IFN-γ use during production impacted BAG6 expression, restoring NKp30 function. Activating NK cells through NKp30 signaling using IFN-γ-Dex may be a promising immunotherapy, especially for NSCLC patients with NKp30-specific defects. These Dex, exosomes derived from dendritic cells, are engineered to elicit immune responses against cancer cells in NSCLC patients [147]. Dex demonstrate significant potential as vaccines for cancer treatment. Yet, their production requires specialized facilities, stringent manufacturing practices, and expensive growth factors and maturation agents [148].

## 7. Application of Exosomes in Drug Delivery

Nanomedicine, an innovative drug delivery technology, has brought about a paradigm shift in the conventional approach to treating severe illnesses, especially cancers [149,150,151,152]. Exosomes, being nano-sized and equipped with cell surface molecules, possess strong interstitial penetration and targeting capabilities. This makes them efficient for targeted delivery of chemotherapeutics. Novel exosome applications involve delivering drugs directly to specific tissues, particularly tumor tissues, showing promise as natural drug carriers [3,131,133,153].

Additionally, exosomes can be engineered to carry ligands that bind to cancer cell receptors, enhancing their specificity and efficacy. Their small size and lipid bilayer enable fusion with target cell membranes, bypassing endosomal–lysosomal pathways and improving drug delivery efficiency. Moreover, modified exosomes can feature specific targeting ligands, boosting their affinity for particular cell types. In cancer treatment, exosomes hold their potential for delivering chemotherapeutic agents into cancer cells [132,146,154,155,156,157,158].

Exosomes have been explored in various studies, showcasing their potential in transporting chemotherapeutics into tumor tissues, thereby reducing tumor growth without observing any adverse effects [126,131,159,160]. In vivo and in vitro analyses have shown drugs such as paclitaxel and doxorubicin encapsulated in exosomes that are able of stimulating therapeutic responses in lung cancer [161]. One study investigated using exosomes to deliver the drug doxorubicin into lung cancer cells. These researchers created nanosomes, which are exosomes loaded with stimuli-responsive nanoparticles, by loading doxorubicin attached to gold nanoparticles onto exosomes, forming NanoDox. They exhibited a slower and more consistent release of Dox, leading to prolonged effectiveness in killing tumor cells. These exosomes exhibited delayed and sustained drug release, inducing cell cycle arrest, apoptosis, and DNA damage [131].

Additionally, exosomes offer a promising avenue for targeted therapies in lung cancer through the delivery of specific miRNAs and siRNAs. Certain miRNAs carried by exosomes, such as miR-200c, miR-193a-3p, and miR-193a-5p, demonstrate the potential to inhibit NSCLC cell behavior, including proliferation, migration, and invasion [87,162]. Additionally, re-expressing miR-451 in docetaxel-resistant lung adenocarcinoma cells shows promise in reversing radioresistance through apoptosis promotion and DNA damage induction [163]. These findings underline miRNAs’ crucial roles in cancer and therapeutic potential.

Moreover, exosomes are identified as valuable therapeutic agents capable of targeting multiple pathways involved in lung cancer progression. For instance, exosomal miRNAs like miR-200a show promise in inhibiting tumor growth and metastasis [164,165]. They combat chemotherapeutic resistance through pathways, such as miR-1 targeting ATG3 [166], and miR-181b targeting Notch2 [167]. These observations position exosomes as potential carriers for enhancing lung cancer treatment. Notably, miR-148b stands out as a tumor suppressor in NSCLC, acting on the MAPK/JNK pathway [168]. As studies continue to unveil the diverse roles of exosomes, further investigation and clinical validation become crucial to fully harness their potential in both lung cancer therapy and monitoring. While caution is advised regarding the use of tumor-derived exosomes for drug delivery due to the potential enhancement of proliferation, additional research is necessary to validate their efficacy and safety.

## 8. Conclusions

In conclusion, exosomes, small extracellular vesicles derived from diverse cell types, represent a burgeoning frontier in the realm of lung cancer diagnosis and therapy. These vesicles boast exceptional attributes that make them highly promising tools in the fight against lung cancer. Serving as potential drug carriers, exosomes exhibit the capacity to transport therapeutic cargo, such as chemotherapeutic drugs, siRNAs, and immune modulators, directly to cancer cells, facilitated by their ability to traverse biological barriers, including the blood–brain barrier. Moreover, engineered exosomes with specific targeting molecules demonstrate a precise delivery mechanism to cancer cells, minimizing off-target effects.

Notably, exosomes also offer invaluable insights for lung cancer diagnosis. Laden with a plethora of molecules, like proteins, nucleic acids, and metabolites, these vesicles serve as a rich source of biomarkers. Analyzing the contents of exosomes, including genetic mutations, microRNAs, and proteins, aids in unraveling the molecular profiles of lung tumors. This analysis supports the early detection, prognostic assessment, and monitoring of treatment responses.

In the realm of targeted drug delivery, exosomes stand as promising carriers, delivering therapeutics precisely to augment treatment efficacy while minimizing adverse effects on healthy tissues. This evolving landscape in exosome research holds the potential to usher in a new era of tailored, efficient, and minimally invasive interventions for managing lung cancer.

Nonetheless, challenges persist in the realm of exosome research. Establishing standardized isolation and purification techniques, along with comprehensive characterization of exosomal cargo, remains pivotal for precision and consistency. Delving deeper into the mechanisms of exosome-mediated drug delivery and refining targeting strategies can amplify the effectiveness and safety of therapies based on exosomes. Additionally, the potential development of advanced technologies capable of detecting and analyzing single exosomes stand poised to revolutionize diagnostic accuracy and offer profound insights into tumor biology, paving the way for personalized treatment strategies.

## Figures and Tables

**Figure 1 biomedicines-12-00123-f001:**
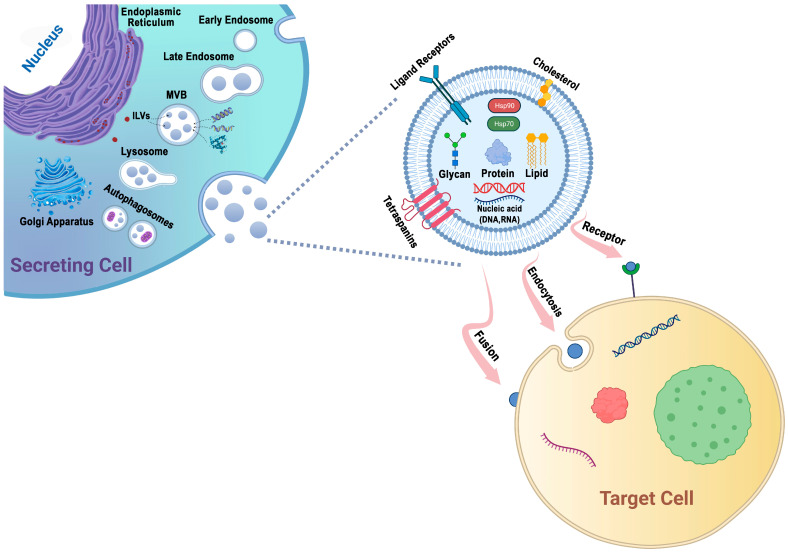
Exosomes are generated through a series of biogenetic processes, starting with the maturation of early endosomes into late endosomes. These late endosomes then transform into MVBs. Within the MVBs, ILVs are formed via the inward budding of the endosomal membrane, which selectively captures proteins, nucleic acids, glycans, and lipids. The MVBs have two main fates: they can fuse with lysosomes for degradation or fuse with the plasma membrane, leading to the release of ILVs as exosomes into the extracellular space. The generation and release of exosomes are influenced by molecules such as tetraspanins and lipids, including cholesterol. These components play important roles in facilitating exosome formation and release. Exosomes possess the capability to adhere to surface receptors present on the target cell, promote endocytosis, or engage in direct fusion with the plasma membrane of the target cell.

**Figure 2 biomedicines-12-00123-f002:**
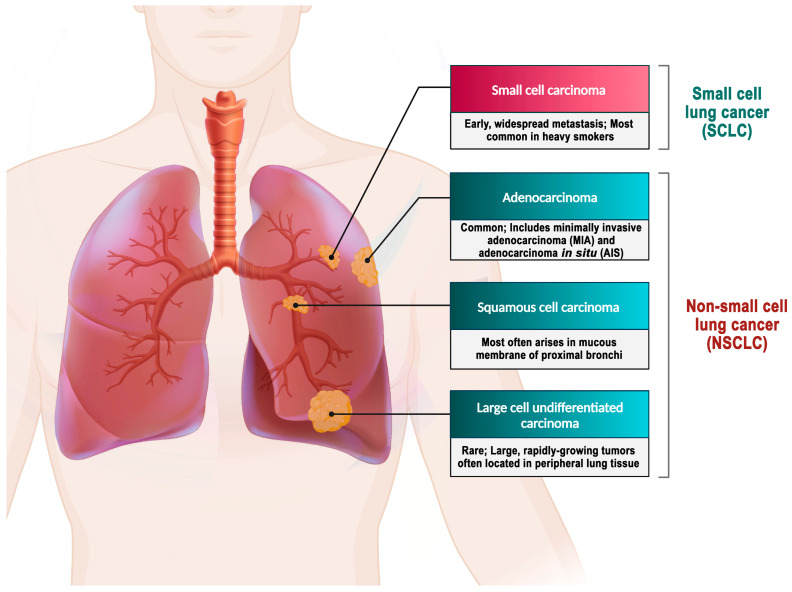
Types of lung cancer. Adenocarcinoma is the most prevalent form of lung cancer, accounting for approximately 40% of all cases [83]. A mutation in the EGFR has been identified as one of the early detectable mutations in lung adenocarcinoma, suggesting its potential as a biomarker for the early detection of this disease [84]. About 25–30% of all lung cancer cases are squamous cell carcinoma, which develops from the premature forms of squamous cells in the airway epithelial cells located in the bronchial tubes at the center of the lungs. This type of NSCLC is highly associated with smoking cigarettes [85]. Large-cell carcinoma is an uncommon form of malignant lung tumor, accounting for 2.9% of lung cancer cases. Despite being categorized as NSCLC, large-cell carcinoma is more severe and malignant than other NSCLCs and shares similar biological characteristics with SCLC. The majority of large-cell carcinoma patients are elderly males who smoke, and their clinical symptoms are not specific [4,86].

**Table 1 biomedicines-12-00123-t001:** Clinical trials on lung cancer diagnosis and treatment via exosome detection since 2015.

Title	Clinicaltrials.gov ID	Intervention/Treatment	Status	Number of Patients
Multicenter clinical research for early diagnosis of lung cancer using blood plasma derived exosome	NCT04529915	Exosome sampling is utilized as a diagnostic test.	in progress	470
Serum exosomal long noncoding rnas as potential biomarkers for lung cancer diagnosis	NCT03830619	Serum samples and clinical features are collected for analysis.	completed	1000
Clinical study of ctDNA and exosome combined detection to identify benign and malignant pulmonary nodules (ctDNA)	NCT04182893	The diagnostic test involves the combined detection of ctDNA and exosomes.	not available	400
Combined diagnosis of ct and exosome in early lung cancer	NCT03542253	Pathological specimens are obtained after surgical treatment to obtain relevant pathological results.	not available	80
Molecular profiling of exosomes in tumor-draining vein of early-staged lung cancer (ExOnSite-Pro)	NCT04939324	Blood samples are collected from 2 sites, the peripheral vein and the tumor-draining vein, on the day of surgery (inclusion) for oncological lung resection. Resected tumor analysis is conducted, and standard clinical and radiological follow-up is performed over a 2-year period.	in progress	30
Improving the early detection of lung cancer by combining exosomal analysis of hypoxia with standard of care imaging (LungExoDETECT)	NCT04629079	This information is not provided.	recruiting	800
Comparision of various biomarkers between peripheral and pulmonary blood	NCT05587114	Frozen blood plasma samples are used for the analysis of various biomarkers.	recruiting	150
Construction of microfluidic exosome chip for diagnosis of lung metastasis of osteosarcoma	NCT05101655	This information is not provided.	completed	60
Circulating exosome rna in lung metastases of primary high-grade osteosarcoma	NCT03108677	Peripheral blood samples of 15 mL are collected.	in progress	90
Circulating and imaging biomarkers to improve lung cancer management and early detection (SMAC-2)	NCT04315753	The role of non-invasive molecular and cellular biomarkers, combined with a radiomic signature, is validated as complementary tools for the early detection of lung cancer using low-dose CT (LDCT). Bioinformatic techniques are employed to integrate and interpret the obtained results.	not available	2000
Non-small cell lung cancer with central nervous system metastas	NCT06026735	Cerebrospinal fluid, blood, and surgically resected brain or meningeal tumor tissue are obtained from non-small cell lung cancer patients with brain/leptomeningeal metastasis for the comprehensive analysis of exosomes. The aim is to investigate whether cancer cell-related substances in these samples can serve as predictive biomarkers for lung cancer metastasis and treatment response.	recruiting	30
Prediction of immunotherapeutic effect of advanced non-small cell lung cancer	NCT04427475	A cohort of 100 patients with stage IV EGFR/ALK wild-type non-small cell lung cancer (excluding squamous-cell carcinoma) receiving first-line anti-PD-1 (pabolizumab) treatment combined with chemotherapy is enrolled. Baseline plasma samples are collected before treatment initiation, and treatment efficacy is evaluated every two treatment cycles. Additional plasma samples are collected at key intervals throughout the patient’s progress. Drug: nafulizumab ◦ nafulizumab	not available	200
Olmutinib trial in t790m (+) nsclc patients detected by liquid biopsy using balf extracellular vesicular dna	NCT03228277	Drug: olmutinib Patients to be provided with olmutinib 800 mg (2 × 400 mg tablets) once daily (QD) Other names: Olita^®^ (MedChemExpress, Monmouth Junction, NJ, USA)	completed	25
The study of exosome EML4-ALK fusion in nsclc clinical diagnosis and dynamic monitoring	NCT04499794	ALK inhibitor treatment is administered to patients with ALK fusion-positive non-small cell lung cancer.	recruiting	75
Clinical research for the consistency analysis of PD-L1 in lung cancer tissue and plasma exosome before and after radiotherapy (RadImm02)	NCT02869685	Radiation: radiotherapy	not available	60

## Data Availability

Not applicable.

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
