# Peer review of "Extracellular Vesicles and Exosomes: Novel Insights and Perspectives on Lung Cancer from Early Detection to Targeted Treatment"

_biomedicines, 2024, doi:10.3390/biomedicines12010123_

Round 1
Reviewer 1 Report
Comments and Suggestions for Authors
The article by Sana Rahimian and colleagues entitled “The Role of Exosomes in Lung Cancer: From Early Detection to Targeted Treatment” is a review about the role of extracellular vesicles in lung cancer. Given the fact that the authors don`t focus on exosomes during the review the title has to be changed and “exosomes” must be replaced by “extracellular vesicles”. In this review you find very often repetitions – some parts are repeated word by word other parts are repeated analogously. Both manners of repletion are not acceptable and furthermore not at all necessary. The authors have to delete all these repetitions. Furthermore, you notice that this review was written by different authors and it is necessary to unify the text, e.g. citation at the end of a sentence or at the beginning of a sentence. Some parts are also in need of improvement in regard to English. In summary, this review needs significant revision and the current version cannot be published in the journal “biomedicines”.
My further major concerns are:
- Very often citations for statements are missing; the authors have to prove all statements with citations; there are so many missing that I cannot mention all here.
- The authors have to use more up to date citations.
- The authors should introduce abbreviation only when these are also used in the following text.
- On the other hand, the authors have to introduce an abbreviation before using it.
- The number of patients included or planned to include in the clinical studies must be added to Table 1.
- It is neither nice nor necessary to read “Cazzoli et al... ….” etc. if somebody is interested in the name of the first author (s)he will find this information in the Reference list. The authors have to rephrase all these parts.
- In Table 2 the clinical trial numbers must be added.
- If the authors state something like “Many articles….” more than two citation are necessary.
- If the authors state something like “In recent researches,….” at least two citation are necessary.
- The Conclusion part is very weak and completely inacceptable. Simply repeating already stated points is far away from being good. The authors have to rewrite this part.
There are a lot of minor concerns that must be addressed too. But this list would be several pages long and I have not the time to write them all done.
Comments on the Quality of English LanguageModerate editing of English language required
Author Response
Reviewer #1:
- The article by Sana Rahimian and colleagues entitled “The Role of Exosomes in Lung Cancer: From Early Detection to Targeted Treatment” is a review about the role of extracellular vesicles in lung cancer. Given the fact that the authors don`t focus on exosomes during the review the title has to be changed and “exosomes” must be replaced by “extracellular vesicles”. In this review you find very often repetitions – some parts are repeated word by word other parts are repeated analogously. Both manners of repletion are not acceptable and furthermore not at all necessary. The authors have to delete all these repetitions. Furthermore, you notice that this review was written by different authors, and it is necessary to unify the text, e.g. citation at the end of a sentence or at the beginning of a sentence. Some parts are also in need of improvement in regard to English. In summary, this review needs significant revision, and the current version cannot be published in the journal “biomedicines”.
Authors respond: We sincerely appreciate your thorough evaluation and the time you have invested in reviewing our manuscript, titled "The Role of Exosomes in Lung Cancer: From Early Detection to Targeted Treatment." We would like to assure you that we have carefully considered your comments and have taken significant steps to address the concerns raised.
We acknowledge your observation regarding the repetitions within the manuscript. We agree that such repetitions are not acceptable and can hinder the readability of the review. During the revision process, we have meticulously reviewed the entire manuscript and have removed all instances of word-for-word and analogous repetitions.
Furthermore, we have addressed the need for unifying the text by standardizing the citation style throughout the manuscript. We have carefully examined the citations and have made the necessary revisions to ensure consistency and clarity.
Lastly, we have focused on improving the overall quality of the English language in the manuscript. We have thoroughly reviewed and edited the text to address any areas that required improvement.
We genuinely appreciate your constructive feedback, as it has been instrumental in guiding our revision process. We are committed to making the necessary revisions to ensure the manuscript meets the high standards set by the journal "Biomedicines."
My further major concerns are:
- Very often citations for statements are missing; the authors have to prove all statements with citations; there are so many missing that I cannot mention all here.
Authors respond: In response to your comment, we have carefully considered your comment regarding missing citations and have taken significant measures to address this concern. We have diligently ensured that all statements are supported by appropriate citations. We have taken the necessary steps to locate and rectify any instances where citations were missing or insufficient. The revised version of the paper now includes some new citations, which have been prominently highlighted in the text.
- The authors have to use more up to date citations.
Authors respond: In response to your comment, we have conducted a thorough assessment of the citations used in the text. While it is true that a few citations before 2009 were included, we would like to emphasize that the majority of our citations are up to date. We have made a concerted effort to ensure that the most recent and relevant research findings are reflected in our paper.
To further address this concern, we have also added some new citations that specifically cover recent developments in the field. These new citations have been prominently highlighted in the text to draw attention to their relevance and timeliness.
- The authors should introduce abbreviation only when these are also used in the following text.
Authors response: In response to your concern, we have thoroughly reviewed the manuscript and made sure that all abbreviations mentioned are appropriately introduced and defined in the text. By adhering to the standard practice in academic writing, we have provided the full form of any abbreviations upon their first mention and subsequently used the abbreviations in a consistent manner throughout the manuscript.
- On the other hand, the authors have to introduce an abbreviation before using it.
Authors response: We have conducted a thorough review of the manuscript and taken careful measures to ensure that all abbreviations are introduced before their first use.
- The number of patients included or planned to include in the clinical studies must be added to Table 1.
Authors response: In response to your comment, we have made the necessary revisions to Table 1 to include the number of patients included or planned to be included in the clinical trials. Additionally, this information is prominently highlighted in the table.
- It is neither nice nor necessary to read “Cazzoli et al... ….” etc. if somebody is interested in the name of the first author (s)he will find this information in the Reference list. The authors have to rephrase all these parts.
Authors response: In response to your comment, we have carefully rephrased all instances where author names were repeatedly mentioned in the text. We genuinely appreciate your attention to this matter, and we believe that the revised version of our paper now effectively addresses your concern regarding the repetitive use of author names.
- In Table 2 the clinical trial numbers must be added.
Authors response: In response to your comment, we have thoroughly revised Table 2 to include the clinical trial numbers. We have carefully ensured that this information is now present in the table, providing readers with the necessary details regarding the clinical trials referenced in our paper. Furthermore, we have highlighted this specific part of the table.
- If the authors state something like “Many articles….” more than two citation are necessary.
Authors response: In response to your comment, we have carefully reviewed the manuscript and have made significant efforts to address this concern. We agree that when making a statement such as "Many articles...", it is essential to provide appropriate citations to support the claim and ensure the credibility of our work. To rectify this issue, we have added more citations to substantiate statements where multiple articles are referenced.
- If the authors state something like “In recent researches,….” at least two citation are necessary.
Authors response: In response to your comment, we have taken steps to address this concern by adding more citations to substantiate statements referencing recent research. By incorporating these additional citations, we have ensured that the claims made in relation to recent research are well-supported and grounded in the existing literature.
- The Conclusion part is very weak and completely inacceptable. Simply repeating already stated points is far away from being good. The authors have to rewrite this part.
Authors response: We acknowledge the importance of a strong and well-crafted Conclusion that effectively summarizes the key findings and insights of our review. In response to your comment, we have thoroughly reworked and revised the Conclusion section of our manuscript. We have taken into account the hints and suggestions you provided regarding the weaknesses in our original Conclusion.
- There are a lot of minor concerns that must be addressed too. But this list would be several pages long and I have not the time to write them all done.
Authors response: Thank you for your thorough evaluation of our manuscript. We understand that there may be numerous minor concerns that need to be addressed, and we value your feedback in helping us further refine our work. We want to assure you that we have made extensive revisions to address the identified issues and have strived to eliminate as many problems as possible. We have carefully considered your insights and utilized them as a valuable resource to identify additional areas that needed attention.
Reviewer 2 Report
Comments and Suggestions for Authors
The manuscript aims to review the role of exosomes in lung cancer. In general, the information is presented in a confusing and highly repetitive way. Key concepts are not highlighted from less relevant information. In addition, a significant part of the text does not focus on lung cancer but is rather general.
1. Currently it is very difficult to distinguish between exosomes and the other EVs. Therefore, unless an unequivocal characterization has been performed in the source literature, the authors should use the term EV instead of exosomes. This is also in agreement with the guidelines of the ISEV.
2. The paragraphs in lines 62-75 and 76-87 should be rearranged. Defining first exosomes and then EVs may be quite confusing.
3. Please, add a reference for the statement in lines 85-87.
4. The information in lines 62 and 110 on the size of exosomes is different. Please, check.
5. Lines 131-151, 188-193, 220-224. The information on the role of ESCRT proteins in the exosome biogenesis is repeated several times.
6. Line 224. The statement is wrong. ESCRT expression is not restricted to the syncytiotrophoblast.
7. The information in lines 386-416 highly overlaps with the information presented in the section 1.
8. The information presented in table 1 describing mostly ongoing and planned clinical trials, which have not delivered data yet, does not belong, in my opinion, to a literature review.
9. The information in lines 453-454 and 458-467 belongs to the main subject of this review. Therefore, the original study (not another review article) needs to be cited and described.
10. Lines 502-509. Several references are missing. The only reference for this part is the Ref. 140, which only describes EGFR.
11. Section 6.2. Most of the information presented between lines 538 and 608 is general and not particularly related to lung cancer. Some of this information was also presented in the introduction.
12. Section 7. The information on miRNAs partially repeats the concepts presented in pages 29 and 30.
Author Response
Reviewer #2:
The manuscript aims to review the role of exosomes in lung cancer. In general, the information is presented in a confusing and highly repetitive way. Key concepts are not highlighted from less relevant information. In addition, a significant part of the text does not focus on lung cancer but is rather general.
Authors respond: Thank you for taking the time to evaluate our manuscript on the role of exosomes in lung cancer.
We acknowledge your comment regarding the confusing and repetitive presentation of information in the manuscript. During the revision process, we have made significant efforts to streamline the content, eliminate unnecessary repetition, and enhance the overall organization of the manuscript.
We also recognize your observation that a significant portion of the text is not specifically focused on lung cancer but rather provides more general information. In our aim to provide a comprehensive review, we included some general information to establish a foundation before delving into the specific role of exosomes in lung cancer. We have revised the text to ensure that the content is more tightly linked to lung cancer and its association with exosomes.
We genuinely appreciate your constructive feedback, as it has guided us in making the necessary improvements to enhance the clarity, organization, and relevance of the manuscript. We are committed to providing a comprehensive and focused review of the role of exosomes in lung cancer.
- Currently it is very difficult to distinguish between exosomes and the other EVs. Therefore, unless an unequivocal characterization has been performed in the source literature, the authors should use the term EV instead of exosomes. This is also in agreement with the guidelines of the ISEV.
Authors respond: Thank you for your valuable feedback on our manuscript. We appreciate your insights regarding the distinction between exosomes and other extracellular vesicles (EVs). We would like to clarify our rationale for using the term "exosomes" throughout the manuscript. Our study specifically focuses on the specific characteristics and roles of exosomes, rather than discussing EVs in a general sense. Given the specialized nature of our research and the specific emphasis on exosomes, we believe that using the term "exosomes" accurately reflects the scope and focus of our study.
- The paragraphs in lines 62-75 and 76-87 should be rearranged. Defining first exosomes and then EVs may be quite confusing.
Authors respond: We want to assure you that we have carefully considered your feedback and made the necessary revisions to address this concern. We have reorganized the paragraphs in the specified sections, ensuring that the definition and explanation of EVs precede the broader discussion of exosomes.
- Please, add a reference for the statement in lines 85-87.
Authors respond: In response to your suggestion, we have carefully reviewed the manuscript and have added a reference to substantiate the statement in question. We have ensured that the added reference aligns with the specific statement made in lines 85-87, providing further evidence and support for the information presented. To draw your attention to the added reference, we have highlighted it in the revised version of the manuscript.
- The information in lines 62 and 110 on the size of exosomes is different. Please, check.
Authors respond: We have thoroughly reviewed the manuscript and have taken immediate action to address this issue. To provide the most reliable and up-to-date information, we have omitted the specific size measurements mentioned in lines 62 and 110. Instead, we have added a new reference that presents more recent findings on exosome size, which we believe to be more reliable and relevant to the current understanding of exosomes. Additionally, we have highlighted the added part related to the size of exosomes.
- Lines 131-151, 188-193, 220-224. The information on the role of ESCRT proteins in the exosome biogenesis is repeated several times.
Authors respond: We want to assure you that we have thoroughly reviewed the manuscript and have made the necessary revisions to eliminate the repetition of information. Specifically, we have removed the redundant information presented in lines 188-193 and lines 220-224, as you have rightly pointed out. However, we have retained the essential information provided in lines 131-151, as it contributes to the overall understanding of the role of ESCRT proteins in exosome biogenesis.
- Line 224. The statement is wrong. ESCRT expression is not restricted to the syncytiotrophoblast.
Authors respond: Thank you for bringing this to our attention and for pointing out that the expression of ESCRT is not restricted to the syncytiotrophoblast. Upon careful consideration of your comment, we have acknowledged the error and have taken immediate action to rectify it. We have revised the manuscript to remove the inaccurate statement, ensuring that the information presented aligns with the current understanding of ESCRT expression.
- The information in lines 386-416 highly overlaps with the information presented in the section 1.
Authors respond: We want to assure you that we have carefully reviewed the content and removed any repetitive information from each section. During this revision process, we have retained the essential and necessary information that best suits the specific purpose and context of each section. We have made sure that the introduction provides a comprehensive overview of the topic, while lines 386-416 focus on specific aspects that require elaboration and further discussion. By carefully selecting and preserving the pertinent information in each section, we have aimed to avoid significant repetitions and maintain the integrity and clarity of the manuscript.
- The information presented in table 1 describing mostly ongoing and planned clinical trials, which have not delivered data yet, does not belong, in my opinion, to a literature review.
Authors respond: By including information about ongoing and planned trials, we aim to fulfill the needs of readers who are genuinely interested in understanding the entire spectrum of research being conducted in this field. It allows them to have a holistic view of the ongoing efforts and potential future directions in clinical research.
- The information in lines 453-454 and 458-467 belongs to the main subject of this review. Therefore, the original study (not another review article) needs to be cited and described.
Authors respond: In the updated version of the manuscript, we have included the appropriate citation to the original study and provided a detailed description of its findings. We have taken significant measures to address this concern and provide a comprehensive and well-supported analysis of the main subject.
- Lines 502-509. Several references are missing. The only reference for this part is the Ref. 140, which only describes EGFR.
Authors respond: Upon careful review, we identified the missing references and have added them to the relevant section of the manuscript. By incorporating these additional references, we have ensured that the information presented is properly backed by relevant and credible sources.
- Section 6.2. Most of the information presented between lines 538 and 608 is general and not particularly related to lung cancer. Some of this information was also presented in the introduction.
Authors respond: In response to your comment, we have made significant revisions to this section to ensure that it contains more specific and lung cancer-related research and information. We have carefully reevaluated the content and removed any extraneous or unrelated information that did not directly contribute to the understanding of the role of exosomes in lung cancer. By refining the section, we have aimed to provide a more targeted and concise discussion that highlights the specific relevance of exosomes in the context of lung cancer.
- Section 7. The information on miRNAs partially repeats the concepts presented in pages 29 and 30.
Authors respond: We would like to clarify that the discussion of miRNA in Section 7 is primarily focused on the therapeutic aspect of exosomes and their potential in drug delivery approaches. On the other hand, the information presented on pages 29 and 30 primarily centers around the diagnostic applications and usage of exosomal miRNA as biomarkers.
Although both sections touch upon the role of miRNA, it is important to note that the context in which they are discussed differs significantly. The therapeutic section explores the potential of utilizing miRNA-loaded exosomes for targeted drug delivery, highlighting their potential as a therapeutic tool. In contrast, the diagnostic section emphasizes the utility of exosomal miRNA as non-invasive biomarkers for early detection and monitoring of lung cancer.
Reviewer 3 Report
Comments and Suggestions for Authors
The authors need to revise the English as suggested and reorganize the article to highlight the role of exosomes in lung cancer. the authors did not mention the rationale for focusing on exosomes, not whole EVs. They also did not talk about oncosomes and other Populations of EVs. A comparison table with references between EVS populations and their role in lung cancer would be great to add.

Comments on the Quality of English LanguageThe authors need to revise the English as suggested and reorganize the article to highlight the role of exosomes in lung cancer. the authors did not mention the rationale for focusing on exosomes, not whole EVs. They also did not talk about oncosomes and other Populations of EVs. A comparison table with references between EVS populations and their role in lung cancer would be great to add.
Author Response
Reviewer #3:
- The authors need to revise the English as suggested and reorganize the article to highlight the role of exosomes in lung cancer. the authors did not mention the rationale for focusing on exosomes, not whole EVs. They also did not talk about oncosomes and other Populations of EVs. A comparison table with references between EVS populations and their role in lung cancer would be great to add.
Authors respond: Firstly, we apologize if it was not explicitly stated in the manuscript why we focused specifically on exosomes rather than whole EVs. The rationale for this choice was to provide a more focused and in-depth analysis of the role of exosomes in lung cancer.
Regarding your comment on oncosomes and other populations of EVs, we understand their importance in the context of lung cancer. However, due to the scope and specific focus of our study, we intentionally omitted a detailed discussion on these specific EV populations.
We appreciate your suggestion of including a comparison table between different EV populations and their roles in lung cancer. While this is a valuable idea, given our specific focus on exosomes, it would not be appropriate for us to include such a table in our manuscript. However, we have taken your suggestion into consideration and have further enhanced the paragraphs and sections to provide a clear and comprehensive understanding of exosomes in the context of lung cancer.
We have also thoroughly revised the manuscript to enhance the English language and ensure clarity and coherence throughout. Your feedback has been invaluable in this regard, and we thank you for bringing these concerns to our attention.
Round 2
Reviewer 1 Report
Comments and Suggestions for Authors
The article by Sana Rahimian and colleagues entitled “The Role of Exosomes in Lung Cancer: From Early Detection to Targeted Treatment” has been improved significantly. It is still a review about the role of extracellular vesicles in lung cancer and given the fact that the authors don`t focus on exosomes during the review the title has to be changed and “exosomes” must be replaced by “extracellular vesicles”.
Only after changing the title this review can be published in the journal “biomedicines”.
Comments on the Quality of English Language
Minor editing of English language required
Reviewer 2 Report
Comments and Suggestions for Authors
The authors have addressed some of my concerns. However, the manuscript still requires substantial improvement. For instance, in some parts, the focus of the manuscript is not fully clear. The manuscript contains too much information which is not related to the main focus of the article (exosomes in lung cancer).
1. Ref. 10 (line 38). The cited article refers to breast cancer (not lung cancer) and should be, therefore, replaced by another reference.
2. Line 53. The reference 15 is not appropriate for this statement. Please, check.
3. Information in lines 67-69 and 77-78 is basically the same.
4. The information on the size of Exosomes (lines 62, 80, 100) is not precise. Difference sizes are indicated in the different parts of the manuscript. I am aware that the data from the literature are not uniform. However, for the sake of clarity, only one range should be indicated throughout the manuscript.
5. Information in lines 65-66 and 113-114 is the same.
6. Section 2. The description of the biogenesis of exosomes is too long (4 ? pages) and needs to be summarized. Tissue-specific mechanisms are described, which are poorly related to lung cancer (the subject of the review).
7. Lines 333 and 334. The statement on exosomes as biomarkers is minimally related to the rest of the content present in the section (focused on exosomes and drug delivery).
8. Most of the information in section 6 (basic information on lung cancer) was already present in the introduction.
9. Line 564. It is not clear, if this is a separate section or if it is also part of section 6.2.
10. The criteria used to summarize cases of miRNAs in EVs as biomarkers in table 2 are not clear. Not all the cases presented in lines 430-493 are summarized in table 2.
11. Line 618. Reference 195 corresponds to a retracted article. Therefore, the reference and the related content should be removed from the manuscript.
12. The conclusion section is quite long and should be summarized. Furthermore, a conclusion should not include new information or new references.
Comments on the Quality of English Language
Minor editing of English Language required.
Round 3
Reviewer 2 Report
Comments and Suggestions for Authors
The authors have addressed all my concerns.
Comments on the Quality of English LanguageMinor editing of English required.